# Cross-Spectral Factor Analysis

Neil M. Gallagher[1,*], Kyle Ulrich[2,*], Austin Talbot[3],
Kafui Dzirasa[1,4], Lawrence Carin[2] and David E. Carlson[5,6]

[1]Department of Neurobiology, [2]Department of Electrical and Computer Engineering, [3]Department of Statistical Science, [4]Department of Psychiatry and Behavioral Sciences, [5]Department of Civil and Environmental Engineering, [6]Department of Biostatistics and Bioinformatics , Duke University
[*]*Contributed equally to this work*
{neil.gallagher,austin.talbot,kafui.dzirasa,
lcarin,david.carlson}@duke.edu

## Abstract

In neuropsychiatric disorders such as schizophrenia or depression, there is often a disruption in the way that regions of the brain synchronize with one another. To facilitate understanding of network-level synchronization between brain regions, we introduce a novel model of multisite low-frequency neural recordings, such as local field potentials (LFPs) and electroencephalograms (EEGs). The proposed model, named Cross-Spectral Factor Analysis (CSFA), breaks the observed signal into factors defined by unique spatio-spectral properties. These properties are granted to the factors via a Gaussian process formulation in a multiple kernel learning framework. In this way, the LFP signals can be mapped to a lower dimensional space in a way that retains information of relevance to neuroscientists. Critically, the factors are *interpretable*. The proposed approach empirically allows similar performance in classifying mouse genotype and behavioral context when compared to commonly used approaches that lack the interpretability of CSFA. We also introduce a semi-supervised approach, termed discriminative CSFA (dCSFA). CSFA and dCSFA provide useful tools for understanding neural dynamics, particularly by aiding in the design of causal follow-up experiments.

## 1 Introduction

Neuropsychiatric disorders (e.g. schizophrenia, autism spectral disorder, etc.) take an enormous toll on our society [16]. In spite of this, the underlying neural causes of many of these diseases are poorly understood and treatments are developing at a slow pace [2]. Many of these disorders have been linked to a disruption of neural dynamics and communication between brain regions [10, 33]. In recent years, tools such as optogenetics [15, 26] have facilitated the direct probing of causal relationships between neural activity in different brain regions and neural disorders [28]. Planning a well-designed experiment to study spatiotemoral dynamics in neural activity can present a challenge due to the high number of design choices, such as which region(s) to stimulate, what neuron types, and what stimulation pattern to use. In this manuscript we explore how a machine learning approach can facilitate the design of these experiments by developing *interpretable* and *predictive* methods. These two qualities are crucial because they allow exploratory experiments to be used more effectively in the design of causal studies.

We explore how to construct a machine learning approach to capture neural dynamics from raw neural data during changing behavioral and state conditions. A body of literature in theoretical and experimental neuroscience has focused on linking synchronized oscillations, which are observable in LFPs and EEGs, to neural computation [18, 24]. Such oscillations are often quantified by

spectral power, coherence, and phase relationships in particular frequency bands; disruption of these relationships has been observed in neuropsychiatric disorders [20, 33]. There are a number of methods for quantifying synchrony between pairs of brain regions based on statistical correlation between recorded activity in those regions [36, 5], but current methods for effectively identifying such patterns on a multi-region network level, such as Independent Component Analysis (ICA), are difficult to transform to actionable hypotheses.

The motivating data considered here are local field potentials (LFPs) recorded from implanted depth electrodes at multiple sites (brain regions). LFPs are believed to reflect the combined local neural activity of hundreds of thousands of neurons [9]. The unique combination of spatial and temporal precision provided by LFPs allows for accurate representation of frequency and phase relationships between activity in different brain regions. Notably, LFPs do not carry the signal precision present in spiking activity from signal neurons; however, LFP signal characteristics are more consistent between animals, meaning that information gleaned from LFPs can be used to understand *population* level effects, just as in fMRI or EEG studies. Our empirical results further demonstrate this phenomenon.

Multi-region LFP recordings produce relatively high-dimensional datasets. Basic statistical tests typically perform poorly in such high dimensional spaces without being directed by prior knowledge due to multiple comparisons, which diminish statistical power [27]. Furthermore, typical multi-site LFP datasets are both "big data" in the sense that there are a large number of high-dimensional measurements and "small data" in the sense that only a few animals are used to represent the entire population. A common approach to address this issue is to describe such data by a small number of factors (e.g. dimensionality reduction), which increases the statistical power when relevant information (e.g. relationship to behavior) is captured in the factors. Many methods for reducing the dimensionality of neural datasets exist [14], but are generally either geared towards spiking data or simple general-purpose methods such as principal components analysis (PCA). Therefore, reducing the dimensionality of multi-channel LFP datasets into a set of *interpretable* factors can facilitate the construction of *testable* hypotheses regarding the role of neural dynamics in brain function.

The end goal of this analysis is not simply to improve predictive performance, but to design meaningful future causal experiments. By identifying functional and interpretable networks, we can form educated hypotheses and design targeted manipulation of neural circuits. This approach has been previously successful in the field of neuroscience [10]. The choice to investigate networks that span large portions of the brain is critical, as this is the scale at which most clinical and scientific *in vivo* interventions are applied. Additionally, decomposing complex signatures of brain activity into contributions from individual functional networks (i.e. factors) allows for models and analyses that are more conceptually and technically tractable.

Here, we introduce a new framework, denoted Cross-Spectral Factor Analysis (CSFA), which is able to accurately represent multi-region neural dynamics in a low-dimensional manifold while retaining interpretability. The model defines a set of factors, each capturing the power, coherence, and phase relationships for a distribution of neural signals. The learned parameters for each factor correspond to an interpretable representation of the network dynamics. Changes in the relative strengths of each factor can relate neural dynamics to desired variables. Empirically, CSFA discovers networks that are highly predictive of response variables (behavioral context and genotype) for recordings from mice undergoing a behavioral paradigm designed to measure an animal's response to a challenging experience. We further show that incorporating response variables in a supervised multi-objective framework can further map relevant information into a smaller set of features, as in [30], potentially increasing statistical power.

## 2   Model Description

Here, we describe a model to extract a low-dimensional "brain state" representation from multi-channel LFP recordings. The states in this model are defined by a set of factors, each of which describes a specific distribution of observable signals in the network. The data is segmented into time windows composed of $N$ observations, equally spaced over time, from $C$ distinct brain regions. We let window $w$ be represented by $\boldsymbol{Y}^w = [\boldsymbol{y}_1^w, \ldots, \boldsymbol{y}_N^w] \in \mathbb{R}^{C \times N}$ (see Fig 1[left]). $N$ is determined by the sampling rate and the duration of the window. The complete dataset is represented by the set $\mathcal{Y} = \{\boldsymbol{Y}^w\}_{w=1,\ldots,W}$. Window lengths are typically chosen to be 1-5 seconds, as this temporal resolution is assumed to be sufficient to capture the broad changes in brain state that we are interested

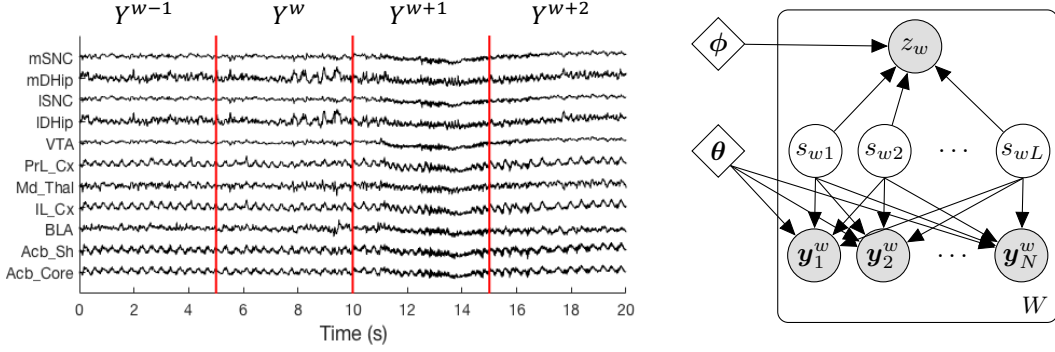

Figure 1: [left] Example of multi-site LFP data from seven brain regions, separated into time windows. [right] Visual description of the parameters of the dCSFA model. $y_c^w$: Signal from channel $c$ in window $w$. $z_w$: Task-relevant side information. $s_{w\ell}$: Score for factor $\ell$ in window $w$. $\theta$: Parameters describing CSFA model. $\phi$: Parameters of side-information classifier. Shaded regions indicate observed variables and clear represent inferred variables.

in. We assume that window durations are short enough to make the signal approximately stationary. This assumption, while only an approximation, is appropriate because we are interested in brain state dynamics that occur on a relatively long time scale (i.e. multiple seconds). Therefore, within a single window of LFP data the observation may be represented by a stationary Gaussian process (GP). It is important to distinguish between signal dynamics, which occur on a time scale of milliseconds, and brain state dynamics, which are assumed to occur over a longer time scale.

In the following, the Cross-Spectral Mixture kernel [34], a key step in the proposed model, is reviewed in Section 2.1. The formulation of the CSFA model is given in Section 2.2. Model inference is discussed in Section 2.3. In Section 2.4, a joint CSFA and classification model called discriminative CSFA (dCSFA) is introduced. Supplemental Section A discusses additional related work. Supplemental Section B gives additional mathematical background on multi-region Gaussian processes. Supplemental Section C offers an alternative formulation of the CSFA model that models the observed signal as the real component of a complex signal. For efficient calculations, computational approximations for the CSFA model are described in Supplemental Section D.

## 2.1 Cross-Spectral Mixture Kernel

Common methods to characterize spectral relationships within and between signal channels are the power-spectral density (PSD) and cross-spectral density (CSD), respectively [29]. A set of multi-channel neural recordings may be characterized by the set of PSDs for each channel and CSDs for each pair of channels, resulting in a quadratic increase in the number of parameters with the number of channels observed. In order to counteract the issues arising from many multiple comparisons, neuroscientists typically preselect channels and frequencies of interest before testing experimental hypotheses about spectral relationships in neural datasets. Instead of directly calculating each of these parameters, we use a modeling approach to estimate the PSDs and CSDs over all channels and frequency bands by using the Cross-Spectral Mixture (CSM) covariance kernel [34]. In this way we effectively reduce the number of parameters required to obtain a good representation of the PSDs and CSDs for a multi-site neural recording.

The CSM multi-output kernel is given by

$$\boldsymbol{K}_{CSM}(t, t'; \mathbf{B}_q, \mu_q, \nu_q) = \text{Real}\left(\sum_{q=1}^{Q} \mathbf{B}_q k_q(t, t'; \mu_q, \nu_q)\right), \tag{1}$$

where the matrix $\boldsymbol{K}_{CSM} \in \mathbb{C}^{C \times C}$. This is the real component of a sum of $Q$ separable kernels. Each of these kernels is given by the combination of a cross-spectral density matrix, $\mathbf{B}_q \in \mathbb{C}^{C \times C}$, and a stationary function of two time points that defines a frequency band, $k_q(\cdot)$. Representing $\tau = t - t'$, as all kernels used here are stationary and depend only on the difference between the two inputs, the frequency band for each spectral kernel is defined by a spectral Gaussian kernel,

$$k_q(\tau; \mu_q, \nu_q) = \exp\left(-\tfrac{1}{2}\nu_q\tau^2 + j\mu_q\tau\right), \tag{2}$$

which is equivalent to a Gaussian distribution in the frequency domain with variance $\nu_q$, centered at $\mu_q$. The matrix $\mathbf{B}_q$ is a positive semi-definite matrix with rank $R$. (Note: The cross-spectral density matrix $\mathbf{B}_q$ is also known as coregionalization matrix in spatial statistics [4]). Keeping $R$ small for the coregionalization matrices ameliorates overfitting by reducing the overall parameter space. This relationship is maintained and $\mathbf{B}_q$ is updated by storing the full matrix as the outer product of a tall matrix with itself:

$$\mathbf{B}_q = \tilde{\mathbf{B}}_q \tilde{\mathbf{B}}_q^{\dagger}, \qquad\qquad \tilde{\mathbf{B}}_q \in C \times R. \qquad (3)$$

Phase coherence between regions is given by the magnitudes of the complex off-diagonal entries in $\mathbf{B}_q$. The phase offset is given by the complex angle of those off-diagonal entries.

## 2.2 Cross-Spectral Factor Analysis

Our proposed model creates a low-dimensional manifold by extending the CSM framework to a multiple kernel learning framework [17]. Let $t_n$ represent the time point of the $n^{th}$ sample in the window and $\boldsymbol{t}$ represent $[t_1, \ldots, t_N]$. Each window of data is modeled as

$$\boldsymbol{y}_n^w = \boldsymbol{f}_w(t_n) + \boldsymbol{\epsilon}_n^w, \qquad\qquad \boldsymbol{\epsilon}_n^w \sim \mathcal{N}(\mathbf{0}, \eta^{-1} I_C), \qquad (4)$$

$$\boldsymbol{F}_w(\boldsymbol{t}) = \sum_{l=1}^{L} s_{wl} \boldsymbol{F}_w^l(\boldsymbol{t}), \qquad\qquad \boldsymbol{F}_w(\boldsymbol{t}) = [\boldsymbol{f}_w(t_1), \ldots, \boldsymbol{f}_w(t_N)], \qquad (5)$$

where $\boldsymbol{F}_w(\boldsymbol{t})$ is represented as a linear combination functions drawn from $L$ latent factors, given by $\{\boldsymbol{F}_w^l(\boldsymbol{t})\}_{l=1}^{L}$. The $l$-th latent function is drawn independently for each task according to

$$\boldsymbol{F}_w^l(\boldsymbol{t}) \sim \mathcal{GP}(\mathbf{0}, \boldsymbol{K}_{CSM}(\cdot; \boldsymbol{\theta}_l)), \qquad (6)$$

where $\boldsymbol{\theta}_l$ is the set of parameters associated with the $l^{th}$ factor (i.e. $\{\mathbf{B}_q^l, \mu_q^l, \nu_q^l\}_{q=1}^{Q}$). The $\mathcal{GP}$ here represents a *multi-output* Gaussian process due to the cross-correlation structure between the brain regions, as in [32]. Additional details on the multi-output Gaussian process formulation can be found in Supplemental Section B.

In CSFA, the latent functions $\{\boldsymbol{F}_w^l(\boldsymbol{t})\}_{l=1}^{L}$ are not the same across windows; rather, the underlying cross-spectral content (power, coherence, and phase) of the signals is shared and the functional instantiation differs from window to window. A marginalization of all latent functions results in a covariance kernel that is a weighted superposition of the kernels for each latent factor, which is given mathematically as

$$\boldsymbol{Y}^w \sim \mathcal{GP}(\mathbf{0}, \boldsymbol{K}_{CSFA}(\cdot; \boldsymbol{\Theta}, w)) \qquad (7)$$

$$\boldsymbol{K}_{CSFA}(\tau; \boldsymbol{\Theta}, w) = \sum_{l=1}^{L} s_{wl}^2 \boldsymbol{K}_{CSM}(\tau; \boldsymbol{\theta}_l) + \eta^{-1} \delta_\tau \boldsymbol{I}_C. \qquad (8)$$

Here, $\boldsymbol{\Theta} = \{\boldsymbol{\theta}_1, \ldots, \boldsymbol{\theta}_L\}$ is the set of parameters associated with all $L$ factors and $\delta_\tau$ represents the Dirac delta function and constructs the additive Gaussian noise. The use of this multi-output GP formulation within the CSFA kernel means that the latent variables can be directly integrated out, facilitating inference.

To address multiplicative non-identifiability, the maximum power in any frequency band is limited for each CSM kernel (i.e. $\max(diag(\boldsymbol{K}_{CSM}(0; \boldsymbol{\theta}_l))) = 1$ for all $l$). In this way, the factor scores squared, $s_{wl}^2$, may now be interpreted approximately as the variance associated with factor $l$ in window $w$.

## 2.3 Inference

A maximum likelihood formulation for the zero-mean Gaussian process given by Eq. 7 is used to learn the factor scores $\{\boldsymbol{s}_w\}_{w=1}^{W}$ and CSM kernel parameters $\boldsymbol{\Theta}$, given the full dataset $\mathcal{Y}$. If we let $\boldsymbol{\Sigma}_{CSFA}^w \in \mathbb{C}^{NC \times NC}$ be the covariance matrix obtained from the kernel $\boldsymbol{K}_{CSFA}(\cdot; \boldsymbol{\Theta}, w)$ evaluated

at time points $t$, we have

$$(\{s_w\}_{w=1}^W, \Theta) = \underset{\{\tilde{s_w}\}_{w=1}^W, \tilde{\Theta}}{\arg\max} \; \mathcal{L}(\mathcal{Y}; \{\tilde{s_w}\}_{w=1}^W, \tilde{\Theta}) \tag{9}$$

$$\mathcal{L}(\mathcal{Y}; \{s_w\}_{w=1}^W, \Theta) = \prod_{w=1}^W \mathcal{N}(\text{vec}(\boldsymbol{Y}^w); \boldsymbol{0}, \boldsymbol{\Sigma}_{CSFA}^w), \tag{10}$$

where $\text{vec}(\cdot)$ gives a column-wise vectorization of its matrix argument, and $W$ is the total number of windows. As is common with many Gaussian processes, an analytic solution to maximize the log-likelihood does not exist. We resort to a batch gradient descent algorithm based on the Adam formulation [22]. Fast calculation of gradients is accomplished via a discrete Fourier transform (DFT) approximation for the CSM kernel [34]. This approximation alters the formulation given in Eq. 7 slightly; the modified form is given in Supplemental Section D. The hyperparameters of the model are the number of factors ($L$), the number of spectral Gaussians per factor ($Q$), the rank of the coregionalization matrix ($R$), and the precision of the additive white noise ($\eta$). In applications where the generative properties of the model are most important, hyperparameters should be chosen using cross-validation based on hold-out log-likelihood. In the results described below, we emphasize the predictive aspects of the model, so hyperparameters are chosen by cross-validating on predictive performance. In order to maximize the generalizability of the model to a population, validation and test sets are composed of data from complete animals/subjects that were not included in the training set.

In all of the results described below, models were trained for 500 Adam iterations, with a learning rate of 0.01 and other learning parameters set to the defaults suggested in [22]. The kernel parameters $\Theta$ were then fixed at their values from the $500^{th}$ iteration and sufficient additional iterations were carried out until the factor scores, $\{s_w\}_{w=1}^W$, reached approximate convergence. Corresponding factor scores are learned for validation and test sets in a similar manner, by initializing the kernel parameters $\Theta$ with those learned from the training set and holding them fixed while learning factor scores to convergence as outlined above. Normalization to address multiplicative identifiability, as described in Section 2.2, was applied to each model after all iterations were completed.

## 2.4  Discriminative CSFA

We often wish to discover factors that are associated with some side information (e.g. behavioral context). More formally, given a set of labels, $\{z_1, \ldots, z_W\}$, we wish to maximize the ability of the factor scores, $\{s_1, \ldots, s_w\}$, to predict the labels. This is accomplished by modifying the objective function to include a second term related to the performance of a classifier that takes the factor scores as regressors. We term this modified model discriminative CSFA, or dCSFA. We choose the cross-entropy error of a simple logistic regression classifier to demonstrate this, giving

$$\{\{s_w\}_{w=1}^W, \Theta\} = \underset{\{\tilde{s_w}\}_{w=1}^W, \tilde{\Theta}}{\arg\max} \; \mathcal{L}(\mathcal{Y}; \{\tilde{s_w}\}_{w=1}^W, \tilde{\Theta}) + \lambda \sum_{w=1}^W \sum_{k=0}^1 1_{z_w=k} \log\left(\frac{\exp(\phi_k \tilde{s_w})}{\sum_k' \exp(\phi_k' \tilde{s_w})}\right). \tag{11}$$

The first term of the RHS of (11) quantifies the generative aspect of how well the model fits the data (the log-likelihood of Section 2.2). The second term is the loss function of classification. Here $\lambda$ is a parameter that controls the relative importance of the classification loss function to the generative likelihood. It is straightforward to include alternative classifiers or side information. For example, when there are multiple classes it is desirable to set the loss function to be the cross entropy loss associated with multinomial logistic regression [23], which only involves modifying the second term of the RHS of (11).

In this dCSFA formulation, $\lambda$ and the other hyperparameters are chosen based on cross-validation of the predictive accuracy of the factors, to produce factors that are predictive as possible in a new dataset from other members of the population. The number of factors included in the classification and corresponding loss function can be limited to a number less than $L$. One application of dCSFA is to find a few factors predictive of side information, embedded in a full set of factors that describe a dataset [30]. In this way, the predictive factors maintain the desirable properties of a generative model, such as robustness to missing regressors. We assume that in many applications of dCSFA, the descriptive properties of the remaining factors matter only in that they provide a larger generative model to embed the discriminative factors in. In applications where the descriptive properties of the

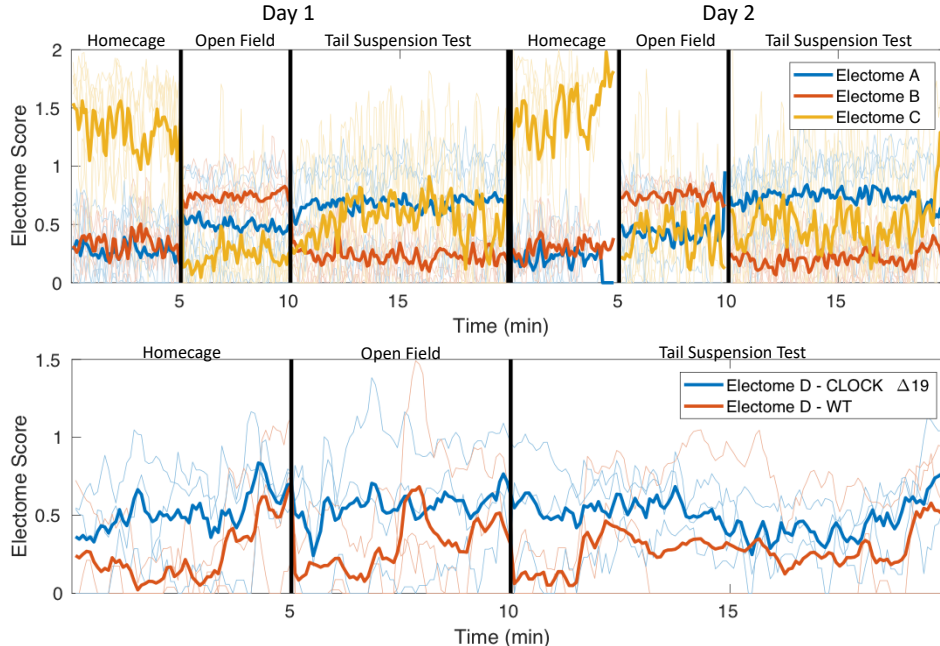

Figure 2: Factor scores learned in two different dCSFA models. Data shown corresponds to the test set described in 3.2. Score trajectories are smoothed over time for visualization. Bold lines give score trajectory averaged over all 6 mice. (top) Scores for three factors that track with behavioral context over a two-day experiment. (bottom) Scores for a single factor that tracks with genotype.

remaining factors are of major importance, hyperparameters can instead be cross-validated using the objective function from (11) applied to data from new members of the population.

## 2.5   Handling Missing Channels

Electrode and surgical failures resulting in unusable data channels are common when collecting the multi-channel LFP datasets that motivate this work. Fortunately, accounting for missing channels is straightforward within the CSFA model by taking advantage of the marginal properties of multivariate Gaussian distributions. This is a standard approach in the Gaussian process literature [31]. Missing channels are handled by marginalizing the missing channel out of the covariance matrix in Eq. 7. This mechanism also allows for the application of CSFA to multiple datasets simultaneously, as long as there is some overlap in the set of regions recorded in each dataset. Similarly, the conditional properties of multivariate Gaussian distributions provide a mechanism for simulating data from missing channels. This is accomplished by finding the conditional covariance matrix for the missing channels given the original matrix (Eq. 8) and the recorded data.

## 3   Results

### 3.1   Synthetic Data

In order to demonstrate that CSFA is capable of accurately representing the true spectral characteristics associated with some dataset, we tested it on a synthetic dataset. The synthetic dataset was simulated from a CSFA model with pre-determined kernel parameters and randomly generated score values at each window. In this way there is a known covariance matrix associated with each window of the dataset. Details of the model used to generate this data are described in Supplemental Section E and Supplemental Table 2. The cross-spectral density was learned for each window of the dataset by training a randomly initialized CSFA model and the KL-divergence compared to the true cross-spectral density was computed. Hyperparameters for the learned CSFA model were chosen to match the model from which the dataset was generated.

A classical issue with many factor analysis approaches, such as probabilistic PCA [7], is the assumption of a constant covariance matrix. To emphasize the point that our method captures dynamics of the covariance structure, we compare the results from CSFA to the KL-divergence from a constant estimate of the covariance matrix over all of the windows, as is assumed in traditional factor analysis approaches. CSFA had an average divergence of $5466.8$ (std. dev. of $49.5$) compared to $7560.2$ (std. dev. of $17.9$) for the mean estimate. These distributions were significantly different (p-value $< 2 \times 10^{-308}$, Wilcoxon rank sum test). This indicates that, on average, CSFA provides a much better estimate of the covariance matrix associated with a window in this synthetic dataset compared to the classical constant covariance assumption.

## 3.2  Mouse Data

We collected a dataset of LFPs recorded from 26 mice from two different genetic backgrounds (14 wild type, 12 CLOCK$\Delta$19). The CLOCK$\Delta$19 line of mice have been proposed as a model of bipolar disorder [35]. There are 20 minutes of recordings for each mouse: 5 minutes while the mouse was in its home cage, 5 minutes during open field exploration, and 10 minutes during a tail suspension test. The tail suspension test is used as an assay of response to a challenging experience [1]. Eleven distinct brain regions were recorded: Nucleus Accumbens Core, Nucleus Accumbens Shell, Basolateral Amygdala, Infralimbic Cortex, Mediodorsal Thalamus, Prelimbic Cortex, Ventral Tegmental Area, Lateral Dorsal Hippocampus, Lateral Substantia Nigra Pars Compacta, Medial Dorsal Hippocampus, and Medial Substantia Nigra Pars Compacta. Following previous applications [34], the window length was set to 5 seconds and data was downsampled to 250 *Hz*.

We learned CSFA and dCSFA models in two separate classification tasks: prediction of animal genotype and of the behavioral context of the recording (i.e. home cage, open field, or tail-suspension test). Three mice of each genotype were held out as a testing set. We used a 5-fold cross-validation approach to select the number of factors, $L$, the number of spectral Gaussians per factor (i.e. factor complexity), $Q$, the rank of the cross-spectral density matrix, $R$, and the additive noise precision, $\eta$. For each validation set, CSFA models were trained for each combination of $L \in \{10, 20, 30\}, Q \in \{3, 5, 8\}, R \in \{1, 2\}, \eta \in \{5, 20\}$, and the model giving the best classification performance on the validation set was selected for testing (see table 1). The hyperparameters above for each dCSFA model were chosen based on the best average performance over all validation sets using CSFA. The parameters for the dCSFA model corresponding to each validation set were initialized from a trained CSFA model for that validation set with the chosen hyperparameters. 3 factors from the CSFA model were chosen to be included in the classifier component of the dCSFA model. For the binary classification task, the 3 factors with the lowest p-value in a Wilcoxon rank-sum test between scores associated with each class were chosen. For the multinomial classification task, a rank-sum test was performed between all pairs of classes, and the 3 factors with the lowest average log p-value were chosen. The $\lambda$ hyperparameter for dCSFA was chosen from $\{1, 0.1, 0.01\}$ based on validation set classification performance.

| Features | Genotype (AUROC) | Behavioral Context (% Accuracy) |
|---|---|---|
| FFT + PCA | 0.632 [0.012] | 85.5 [0.2] |
| Welch + PCA | **0.922 [0.013]** | **87.5 [1.5]** |
| CSFA | 0.685 [0.067] | 82.8 [0.9] |
| dCSFA | 0.731 [0.064] | 83.1 [0.6] |
| dCSFA-3 | **0.741 [0.099]** | **70.7 [1.9]** |
| Welch + PCA-3 | 0.528 [0.045] | 54.7 [0.4] |

Table 1: Classification performance. For genotype, logistic regression with an L1 regularization penalty was used. For behavioral context, multinomial logistic regression with an L2 penalty was used. All results are reported as a mean, with standard error included in brackets. FFT+PCA: PCA applied to magnitude of FFT. Welch+PCA: PCA applied to Welch's estimated spectral densities. CSFA: CSFA factor scores. dCSFA: All dCSFA factor scores. dCSFA-3: Scores for 3 discriminative dCSFA factors. Welch + PCA-3: PCA applied to estimated spectral densities; 3 components selected using the criteria described in 3.2.

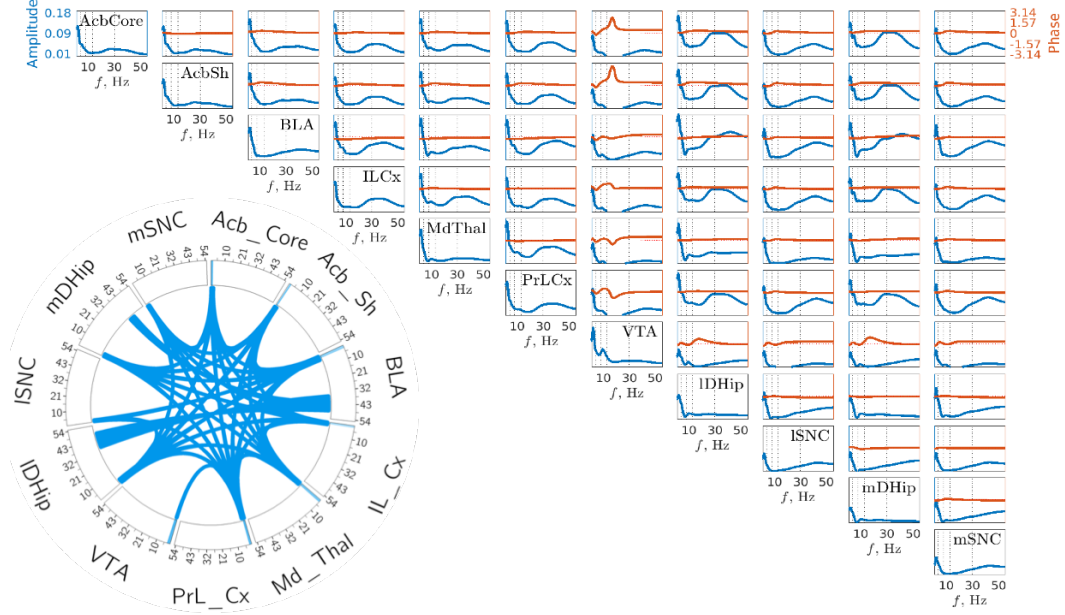

Figure 3: Visual representations of a dCSFA factor. [right] Relative power-spectral (diagonal) and cross-spectral (off-diagonal) densities associated with the covariance function defining a single factor. Amplitude reported for each frequency within a power or cross-spectral density is normalized relative to the total sum of powers or coherences, respectively, at that frequency for all factors. [left] Simplified representation of the same factor. Each 'wedge' corresponds to a single brain region. Colored regions along the 'hub' of the circle represent frequency bands with significant power within that corresponding region. Colored 'spokes' represent frequency bands with significant coherence between the corresponding pair of regions.

We compare our CSFA and dCSFA models to two-stage modeling approaches that are representative of techniques commonly used in the analysis of neural oscillation data [21]. Each of these approaches begins with a method for estimating the spectral content of a signal, followed by PCA to reduce dimensionality (see Supplemental Section F for details). CSFA models were trained as described in Section 2.3; dCSFA models were initialized as reported above and trained for an additional 500 iterations. Figure 2 demonstrates that the predictive features learned from dCSFA clearly track the different behavioral paradigms. If we constrain our classifier to use only a few of the learned features, dCSFA features significantly outperform features from the best comparison method. Compressing relevant predictive information into only a handful of factors here is desirable for a number of reasons; it reduces the necessary number of statistical tests for testing hypotheses and also offers a more interpretable situation for neuroscientists. The dCSFA factor that is most strongly associated with genotype is visualized in Figure 3.

### 3.3 Visualization

The models generated by CSFA are easily visualized and interpreted in a way that allows neuroscientists to generate testable hypotheses related to brain network dynamics. Figure 3 shows one way to visualize the latent factors produced by CSFA. The upper-right section shows the power and cross-spectra associated with the CSM kernel from a single factor. Together these plots define a distribution of multi-channel signals that are described by this one factor. Plots along the diagonal give power spectra for each of the 11 brain regions included in the dataset. The off diagonal plots show the cross spectra with the associated phase offset in orange. The phase offset implies that oscillations may originate in one region and travel to another, given the assumption that another (observed or unobserved) region is not responsible for the observed phase offset. These assumptions are not true in general, so we emphasize that their use is in hypothesis generation.

The circular plot on the bottom-left of Figure 3 visualizes the same factor in an alternative concise way. Around the edge of the circle are the names of the brain regions in the data set and a range of frequencies modeled for each region. Colored bands along the outside of the circle indicate that spectral power in the corresponding region and frequency bands is above a threshold value. Similarly, lines connecting one region to another indicate that the coherence between the two regions is above the same threshold value at the corresponding frequency band. Given the assumption that coherence implies communication between brain regions [5], this plot quickly shows which brain regions are believed to be communicating and at what frequency band in each functional network.

## 4    Discussion and Conclusion

Multi-channel LFP datasets have enormous potential for describing brain network dynamics at the level of individual regions. The dynamic nature and high-dimensionality of such datasets makes direct interpretation quite difficult. In order to take advantage of the information in such datasets, techniques for simplifying and detecting patterns in this context are necessary. Currently available techniques for simplifying these types of high dimensional datasets into a manageable size (e.g. ICA, PCA) generally do not offer sufficient insight into the types of questions that neuroscientists are interested in. More specifically, there is evidence that neural networks produce oscillatory patterns in LFPs as signatures of network activation [18]. Methods such as CSFA, which identify and interpret these signatures at a network level, are needed to form reasonable and testable hypotheses about the dynamics of whole-brain networks. In this work, we show that CSFA detects signatures of multi-region network activity that explain variables of interest to neuroscientists (i.e. animal genotype, behavioral context).

The proposed CSFA model explicitly targets known relationships of LFP data to map the high-dimensional data to a low-dimensional set of features. In direct contrast to many other dimensionality reduction methods, each factor maintains a high degree of interpretability, particularly in neuroscience applications. We emphasize that CSFA captures both spectral power and coherence across brain regions, both of which have been associated with neural information processing within the brain [19]. It is important to note that this model finds temporal precedence in observed signals, rather than true causality; there are many examples where temporal precedence does not imply true causation. Therefore, we emphasize that CSFA facilitates the generation of testable hypothesis rather than demonstrating causal relationships by itself. In addition, CSFA can suggest ways of manipulating network dynamics in order to directly test their role in mental processes. Such experiments might involve closed-loop stimulation using optogenetic or transcranial magnetic stimulation to manipulate the complex temporal dynamics of neural activity captured by the learned factors.

Future work will focus on making these approaches broadly applicable, computationally efficient, and reliable. It is worth noting that CSFA describes the full-cross spectral density of the data, but that there are additional signal characteristics of interest to neuroscientists that are not described, such as cross-frequency coupling [24]; another possible area of future work is the development of additional kernel formulations that could capture these additional signal characteristics. CSFA will also be generalized to include other measurement modalities (e.g. neural spiking, fMRI) to create joint generative models.

In summary, we believe that CSFA fulfills three important criteria: 1. It consolidates high-dimensional data into an easily interpretable low-dimensional space. 2. It adequately represents the raw observed data. 3. It retains information from the original dataset that is relevant to neuroscience researchers. All three of these characteristics are necessary to enable neuroscience researchers to generate trustworthy hypotheses about a network-level brain dynamics.

## Acknowledgements

In working on this project L.C. received funding from the DARPA HIST program; K.D., L.C., and D.C. received funding from the National Institutes of Health by grant R01MH099192-05S2; K.D received funding from the W.M. Keck Foundation.

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
