[Supplementary Material]

## A Additional related work

The goal of this model is to design a general-purpose method for analysis of neural oscillations, specifically an interpretable generative process via a low-dimensional embedding. In machine learning, low-dimensional representations of the data are often used to capture underlying correlated structures, such as topic models [8], state-space models [6], dictionary learning [40], and deep convolutional factor analysis [12]. Due to the inherent non-linearities of oscillatory time-series, non-linear functions are convenient choices to represent this quasi-periodic data. Gaussian processes [31] provide a concise framework for placing prior distributions over these latent functions, and are increasingly being used to represent expressive features through the covariance kernel, such as deep architectures [13], convolutional factor analysis [25], or encoding arbitrary spectral densities via a Gaussian mixture [25]. This latter kernel is known as the spectral mixture (SM) covariance kernel [39], and a recent multi-task [11] extension of the SM kernel, known as the cross-spectral mixture (CSM) kernel [34], encodes the full cross-spectral density between multiple output observations within the GP framework. The CSM kernel was illustrated as an excellent tool for representing multi-output LFP data [34].

## B Multi-output Gaussian processes

A multi-output regression task includes observations from $C$ output channels, $\boldsymbol{Y} = [\boldsymbol{y}_1, \ldots, \boldsymbol{y}_N] \in \mathbb{R}^{C \times N}$ at $N$ time points. The data are modeled as

$$\boldsymbol{y}_n = \boldsymbol{f}_w(t_n). \tag{12}$$

A multi-output Gaussian process [31, 3] places a prior distribution over the latent function, given by

$$\boldsymbol{f}(\cdot) \sim \mathcal{GP}(\boldsymbol{m}(\cdot), \boldsymbol{K}(\cdot, \cdot; \boldsymbol{\theta})). \tag{13}$$

where the Gaussian process is defined by the mean function $\boldsymbol{m}(t) \in \mathbb{R}^C$, and the covariance function $\boldsymbol{K}(t, t'; \boldsymbol{\theta}) \in \mathbb{R}^{C \times C}$. The covariance function defines how signal in $\boldsymbol{f}(\cdot)$ covaries over channels and time points, such that $K^{c,c'}(t_n, t_{n'}; \boldsymbol{\theta}) \triangleq \operatorname{cov}(f_c(t_n), f_{c'}(t_{n'}))$. The mean function is often set to equal $\boldsymbol{0}$. For any set of $N$ points $\boldsymbol{t} = [t_1, \ldots, t_N]$, the values of the function $\boldsymbol{f}$ are drawn from a multivariate normal distribution defined by the mean vector, $\boldsymbol{M}(\boldsymbol{t}) = [\boldsymbol{m}(t_1); \ldots; \boldsymbol{m}(t_N)] \in \mathbb{R}^{NC}$, and the covariance matrix, $\Sigma_K(\boldsymbol{t}; \boldsymbol{\theta}) \in \mathbb{R}^{NC \times NC}$. The covariance matrix $\Sigma_K$ is related to the covariance function $\boldsymbol{K}$ in the following way:

$$\Sigma_K(\boldsymbol{t}; \boldsymbol{\theta}) = \begin{bmatrix} \boldsymbol{K}(t_1, t_1; \boldsymbol{\theta}) & \cdots & \boldsymbol{K}(t_1, t_N; \boldsymbol{\theta}) \\ \vdots & \ddots & \vdots \\ \boldsymbol{K}(t_N, t_1; \boldsymbol{\theta}) & \cdots & \boldsymbol{K}(t_N, t_N; \boldsymbol{\theta}) \end{bmatrix}$$

The parameters $\boldsymbol{\theta}$ may be optimized to fit these observations by maximizing the marginal likelihood

$$\boldsymbol{\theta}^* = \underset{\boldsymbol{\theta}}{\arg\max} \ \log p(\boldsymbol{Y}|\boldsymbol{t}, \boldsymbol{\theta}), \qquad p(\boldsymbol{Y}|\boldsymbol{t}, \boldsymbol{\theta}) = \mathcal{N}\left(\operatorname{vec}(\boldsymbol{Y}); \boldsymbol{M}(\boldsymbol{t}), \Sigma_K(\boldsymbol{t}; \boldsymbol{\theta})\right), \tag{14}$$

where $\operatorname{vec}(\cdot)$ is a column-wise vectorization of its matrix-valued argument. The form of the covariance kernel constrains the types of posterior functions that may be represented by the Gaussian process. Recently, expressive covariance kernels have been explored [39, 38, 34] that are capable of representing any stationary kernel while treating $\boldsymbol{\theta}$ as expressive features of interest extracted from the model.

## C Complex Normal Formulation

Let the observed data for a single window be $\boldsymbol{Y} = [\boldsymbol{y}_1, \ldots, \boldsymbol{y}_N]$; we drop window indexing here for simplicity. From Eq. 7 we model the data as originating from a multivariate normal distribution.

$$\boldsymbol{Y} \sim \mathcal{N}(\boldsymbol{0}, \boldsymbol{K}_{CSFA}(\boldsymbol{t}, \boldsymbol{t}; \boldsymbol{\Theta})). \tag{15}$$

We can consider the observed data, $\boldsymbol{Y}$, to be the real portion of a complex signal, such that

$$\tilde{\boldsymbol{Y}} = \tilde{\boldsymbol{Y}}^r + j\tilde{\boldsymbol{Y}}^i, \qquad \tilde{\boldsymbol{Y}}^r = \boldsymbol{Y}, \qquad \{\tilde{\boldsymbol{Y}}^r, \tilde{\boldsymbol{Y}}^i\} \in \mathbb{R}^{CN}, \tag{16}$$

where superscript $r$ and $i$ correspond to the real and imaginary components of a vector argument and $j$ is the imaginary number. In such as case, we can represent the full complex vector $\tilde{Y}$ as a arising from a multivariate circularly symmetric complex normal distribution.

$$\tilde{Y} \sim \mathcal{CN}(\mathbf{0}, 2\tilde{K}_{CSFA}(\boldsymbol{t}, \boldsymbol{t}; \boldsymbol{\Theta})) \tag{17}$$

$$\tilde{K}_{CSFA}(\boldsymbol{t}, \boldsymbol{t}; \boldsymbol{\Theta}) = \sum_{l=1}^{L} s_{wl}^2 \tilde{K}_{CSM}(\boldsymbol{t}, \boldsymbol{t}; \boldsymbol{\theta}_l) + \eta^{-1}(\boldsymbol{I}_N \otimes \boldsymbol{I}_C) \tag{18}$$

$$\tilde{K}_{CSM}(\boldsymbol{t}, \boldsymbol{t}; \boldsymbol{\theta}_l) = \sum_{q=1}^{Q} \mathbf{B}_q \otimes k_q(\boldsymbol{t}, \boldsymbol{t}; \mu_q, \nu_q), \tag{19}$$

where $\otimes$ denotes the Kronecker product. Note that the only differences between this formulation and that given in the main text are that $\tilde{K}_{CSM}$ equal to the full complex value of the sum in Eq. 19 and that the covariance matrix in Eq. 17 is multiplied by a factor of 2.

## D   DFT Approximation

The computational costs associated with calculating gradients in a CSFA model with a large number of parameters can be quite high, due to the fact that a matrix inversion is necessary for gradient calculation (see [31]). As originally described in [34], the computational costs of the CSM model can be significantly decreased by approximating the covariance matrices associated with all spectral Gaussian kernels (see Sec. 2.1) as circulant matrices. We let $K_{ql}$ be the covariance matrix associated with the $q^{th}$ spectral Gaussian kernel in the $l^{th}$ factor at time points $\boldsymbol{t}$. By definition, $K_{ql}$ is a symmetric Toeplitz matrix, such that it is uniquely identified by its first column, $\boldsymbol{c}$. We get a circulant matrix approximation of $K_{ql}$ by generating a new first column, $\tilde{\boldsymbol{c}}$, by reflecting the first $\lfloor \frac{N}{2} + 1 \rfloor$ elements of $\boldsymbol{c}$ to the last $\lceil \frac{N}{2} + 1 \rceil$. The resulting matrix, $\tilde{K}_{ql}$, is diagonalizable by the discrete Fourier transform (DFT) in the following way. Letting $\boldsymbol{U}$ be the $N \times N$ unitary DFT matrix, and $\delta$ be the sampling period associated with time points $\boldsymbol{t}$, we have

$$\boldsymbol{U}^{\dagger} \tilde{K}_{ql} \boldsymbol{U} = \boldsymbol{\Lambda}_{\tilde{K}_{ql}} = \text{diag}(\delta^{-1} S(\boldsymbol{\omega})), \tag{20}$$

where $\boldsymbol{\omega}$ is the vector of frequencies corresponding to the DFT transformation over $\boldsymbol{t}$, and $S(\cdot)$ is the power spectral density associated with $\tilde{\boldsymbol{c}}$. In this way, we only need to perform computation on the diagonal matrix $\boldsymbol{\Lambda}_{\tilde{K}_{ql}}$, rather than on the full matrix $K_{ql}$. As the sampling rate and window length approach infinity there is no error in approximating $K_{ql}$ with $\tilde{K}_{ql}$ [34], due to sufficient resolution of the DFT frequency bins.

CSFA can take advantage of the same approximation, when formulated in the following way. Letting $\boldsymbol{Z}$ denote the DFT of the observed data, we have

$$\boldsymbol{Z} \sim \mathcal{CN}(\mathbf{0}, 2\boldsymbol{\Sigma}(\boldsymbol{\omega}, \boldsymbol{\omega}; \boldsymbol{\Theta})) \tag{21}$$

$$\boldsymbol{\Sigma}(\boldsymbol{\omega}, \boldsymbol{\omega}; \boldsymbol{\Theta}) = (\boldsymbol{I}_C \otimes \boldsymbol{U})^{\dagger} \tilde{K}_{CSFA}(\boldsymbol{t}, \boldsymbol{t}; \boldsymbol{\Theta})(\boldsymbol{I}_C \otimes \boldsymbol{U}) \tag{22}$$

$$\approx \eta^{-1}(\boldsymbol{I}_N \otimes \boldsymbol{I}_C) + \sum_{l=1}^{L} s_l^2 \sum_{q=1}^{Q} \mathbf{B}_{ql} \otimes \boldsymbol{\Lambda}_{\tilde{K}_{ql}}(\boldsymbol{\omega}; \mu_{ql}, \nu_{ql}). \tag{23}$$

This approximation results in a covariance matrix, $\boldsymbol{\Sigma}$, that is block diagonal, significantly reducing the cost of the matrix inversion necessary for computing gradients.

## E   Artificial Dataset Parameterization

To generate our synthetic dataset we generated random draws from a CSFA model with 4 channels and 3 latent factors, each factor containing a single spectral Gaussian component. The parameters for the original CSFA model from which the data were generated are given in Table 2. We simulated 5000 time windows of 5s sampled at 500Hz. For each window two of the scores were nonzero. The non-zero scores were independently drawn from a uniform distribution, then normalized such that the sum of squared scores for each window added to one. This normalization scheme gives approximately unit variance for the signal in each window.

| Factor | Spectral Gaussian Mean ($Hz$) | Spectral Gaussian Variance ($Hz^2$) | Channel Weights | Channel Shifts |
|---|---|---|---|---|
| 1 | 6 | 1 | $[e^{2.25}\, e^2\, 0\, 0]$ | $[0\, 0\, 0\, -\pi/2]$ |
| 2 | 6 | 1 | $[0\, 0\, e^2\, e^{1.75}]$ | $[0\, 0\, 0\, \pi/2]$ |
| 3 | 10 | 1 | $[e\, e\, e\, e]$ | $[0\, \pi/4\, \pi/2\, 3\pi/4]$ |

Table 2: Parameter values for simulated CSFA dataset

# F   Comparison Methods

To provide a comparison of our method we describe two alternative approaches to classifying side information using the power and cross spectra. Just like CSFA, we divide the recording in to a series of time windows. In the first comparison method, the true power spectrum for each channel is simply estimated by taking the magnitude of the signal in the Fourier domain. In the second method, we estimate the power spectra for each channel, and coherence for each pair of channels, independently using Welch's method [37], a non-parametric method for obtaining a denoised estimate of the power spectra. Letting $F$ equal the number of frequencies used and $C$ equal the number of channels, this gives us a high dimensional ($F * (C + 1) * C/2$) representation of the brain dynamics for each window (ignoring directionality). This dimensionality is reduced for both comparison methods using PCA. The number of factors used in each model was chosen from among {10,20,30} using the same cross-validation scheme as described in section 2.3 for the CSFA hyperparameters.

An advantage of these methods is that the computational cost is lower than CSFA. However, these methods lose the interpretability of CSFA. First, there are no constraints between the estimates (variations in cross spectra are independent from the other channels). The factors are not sparsified in any way, which means that every factor involves every frequency in every channel. This means that interpreting the factors requires interpreting a combination of every frequency and coherence. This contrasts with CSFA where, by using parametric spectral Gaussian distributions and coregionalization matrices, the few parameters have strong significance.

# G   Code Repository

A public repository of code for running the CSFA and dCSFA models in MATLAB is available at

```
https://github.com/neil-gallagher/CSFA
```

.