[Reviews · NeurIPS 2017]

Reviewer 1



The authors propose a method called Cross-Spectral Factor Analysis that decomposes obserfved local field potentials (LFPs) into interacting functional connectomes, called Electomes defined by differing spatiotemporal properties. They then use Gaussian process theory to model interactions between these coherence patterns in a way that may reveal dynamic interaction of the different regions of the brain at mesoscale. This approach is interesting in that it provides a way to rigorously model differing regions LFPs at a scale not readily accessible by certain other modalities such as fMRI or EEG. The presentation is generally clear and the overall approach is well motivated with good premise. The approach is fairly standard from a modeling and learning perspective, although the authors have good insight into applicability of these techniques to simulataneous LFP analysis. The authors do no offer any real interpretation of the results of the method, and perhaps this is natural given that no effective ground truth is known. At a minimum it would seem that more detail could be given on the stability or robustness of the derived Electomes and this should be expanded particularly for publication.

Reviewer 2



The paper considers the problem of performing dimensionality reduction for Local Field Potential (LFP) signals that are often used in the field of neuroscience. The authors propose a factor analysis approach (CSFA), where the latent variables are stationary Gaussian Processes that are able to capture correlations between different channels. The parameters of the factor analysis (i.e. factor scores s_w, for w=1...W) can then be optimised along with the kernel parameters \Theta, by maximising the marginal likelihood. For that purpose the authors employ a resilient backpropagation approach. A discriminative approach is also presented (dCSFA), where the goal is to optimise a heuristic that relies on both the GP marginal likelihood and the loss function for the classification. Although I feel that the findings of the paper could be valuable for the field of neuroscience, I think that some claims regarding the underlying theory are not sufficiently supported. Most importantly, it is not clear why the maximisation of the likelihood in Equation (6) would not be prone to overfitting. It is claimed that "the low-rank structures on the coregionalization matrices dramatically ameliorate overfitting", but it is not clear how this low rank is imposed. Minor Comments: The naming of the variables in the equations is not very consistent, and that could create confusion. For example: Equation (2): \tau is not introduced line 124: C is not introduced (the number of channels, I presume) y_w is presented as a function of either x or t (since we have time-series, having t everywhere would make sense)

Reviewer 3



The authors propose a factor analysis method called CSFA for modelling LFP data, which is a generative model with the specific assumption that factors, called Elecotomes, are sampled from Gaussian processes with cross-spectral mixture kernel. The generative model is straightforward use of CSM, and the estimation is apparently also a known form (resilient back prop; I never heard of it before). I do like the dCSFA formulation. The proposed method focuses on spectral power and phase relationship across regions, and is claimed to bring both better interpretability and higher predictive power. They also extend CSFA to be discriminative to side information such as genetic and behavioral data by incorporating logistic loss (or the likes of it). The authors applied the proposed method to synthetic and real mouse data, and compared it with PCA. == issues == === Page limit === This paper is 10 pages. (references start at page 8 and goes to page 10) === Section 2.1 === The authors claim that reduced dimensionality increases the power of hypothesis testing. In general, this is not true. Any dimensionality-reduction decision is made implicitly upon a hypothesis of dimensionality. The hypothesis testing performed after reduction is conditional on it, making the problem more subtle than presented. === Section 2.5 === lambda in Eq(7) is chosen based on cross-validation of the predictive accuracy. Why was it not chosen by cross-validation of the objective function itself? The authors should report its value along the accuracy in the result section. Would it be extremely large to make the first likelihood term ineffective? === FA or PCA === Eq 3 shows that the noise is constant across the diagonal. The main difference between FA and PCA is allowing each dimension of signal to have different amount of noise. This noise formulation is closer to PCA, isn't it? But then I was confused reading sec 3.2. What's the covariance matrix of interest in sec 3.2? Over time? space? factors? frequencies? === "Causal" === The word 'causal' frequently appears in the first 2 pages of the text, setting high expectations. However, the method itself does not seem to have any explanatory power to discover causal structure or inherent causal hypothesis. Or is the reader supposed to get it from the phase plots? === comparison with baseline PCA in frequency domain === How was the window of data converted to the "Fourier domain"? Was it simply a DFT or was a more sophisticated spectrum estimator used, e.g., multi-taper methods. I ask this because raw FFT/DFT is a bad estimator for the spectrum (very high variance regardless of the length of time window). Neuroscientists often use packages that support such estimation (e.g., MATLAB package chronux). It would not be fair if those are not taken into account. === averaging over 6 mice === Was the CSFA/dCSFA for Fig 3 computed to produce common factor across 6 mice? What justifies such approach? == minor == - L 230, 231: absolute KL div numbers are not useful. At least provide standard error of the estimator. (KL div is often very hard to estimate!!) - Subsection 2.1 & 2.3.1 reads like filler texts. (not much content) - L 124: $s\tilde B_q$? what's $s$ here? - L 228: to to - overall writing can be improved